# Rapid Degradation of Chlortetracycline Using Hydrodynamic Cavitation with Hydrogen Peroxide

**DOI:** 10.3390/ijerph19074167

**Published:** 2022-03-31

**Authors:** Chen Meng, Min Meng, Xun Sun, Congcong Gu, Huiyun Zou, Xuewen Li

**Affiliations:** 1Department of Environment and Health, School of Public Health, Cheeloo College of Medicine, Shandong University, Jinan 250012, China; 15932127710@163.com (C.M.); mengm@sdu.edu.cn (M.M.); gcc18826107751@163.com (C.G.); zouhuiyunsdu@163.com (H.Z.); 2Key Laboratory of High Efficiency and Clean Mechanical Manufacture, Ministry of Education, School of Mechanical Engineering, Shandong University, Jinan 250061, China; xunsun@sdu.edu.cn

**Keywords:** hydrodynamic cavitation, chlortetracycline, Venturi, hydrogen peroxide, degradation mechanism, wastewater treatment

## Abstract

Chlortetracycline (CTC), which has been frequently detected in surface water, is generated primarily by the discharge of high-concentration CTC wastewater from pharmaceutical and livestock plants. The development of effective CTC degradation technology is critical. In this study, the extent of CTC degradation at 80 mg/L was investigated by combining hydrodynamic cavitation (HC) and hydrogen peroxide (H_2_O_2_). The results indicate degradation ratios of 88.7% and 93.8% at 5 and 30 min, respectively. Furthermore, the possible mechanisms of CTC degradation were determined via HPLC-MS. The CTC degradation pathways include ring openings, C–N bond cleavage, demethylation, dehydroxylation, and desaturation in the sole system of HC, and a series of additional reactions, such as glycine conjugation and the cleavage of C–C double bonds, occurs in the binary system of HC + H_2_O_2_. Nevertheless, the treated water poses ecological risks and cannot be directly discharged into the environment. Therefore, HC + H_2_O_2_ treatment may be a rapid and effective primary method for the degradation of high-concentration CTC in pharmaceutical factories.

## 1. Introduction

As a broad-spectrum antibiotic, tetracyclines (TCs) have been widely applied for therapeutics and growth promotion in the livestock industry, and high residual concentrations are often detected in pharmaceutical wastewater. The concentration of TC in pharmaceutical factory wastewater remains relatively high, i.e., up to 180 mg/L, after flocculation treatment [1]. Moreover, a high concentration of TCs (61.0 ± 12.2 mg/L) was detected in the influent of a pharmaceutical wastewater treatment plant, in which chlortetracycline (CTC) was present [2]. CTC was first discovered in TCs and has been frequently detected in pharmaceutical and livestock plant wastewater. They are typically drained into surface water [3]. The residue of CTC can cause several adverse effects, such as an increase in the possibility of TC resistance genes [4] and the inhibition of microbial activity and growth [5], which have garnered increasing public attention.

CTC exhibits the structure of a four-ring system which has multiple substituent groups including O- and N-containing functional groups (e.g., -OH and -NH2) on the molecular skeleton, as shown in Figure 1. Various methods have been developed to eliminate CTC; however, they are low in efficiency, costly, and time consuming. For example, traditional wastewater treatment methods (such as membrane filtration and adsorption) can separate antibiotics from wastewater instead of completely degrading them, which results in secondary pollution [6]. Photocatalysts exhibit a few limitations, such as a large bandgap, easy aggregation, small surface area, and instability [7]. Electrocatalytic oxidation ensures a higher degradation (99.6%) of CTC under optimal conditions; nevertheless, the process time is lengthy [8]. Therefore, a rapid and effective CTC treatment technology must be developed.

Recently, hydrodynamic cavitation (HC) has indicated considerable application prospects in the field of sewage treatment [9], owing to numerous advantages, such as its small scale, high energy efficiency, and low energy consumption [10,11,12,13]. HC occurs when a fluid passes through a contractile-divergent structure in an orifice plate or a Venturi [14,15,16]. In the contraction zone, the pressure decreases as the flow rate increases, thereby forming a cavity. Then, in the divergence zone downstream, the pressure is restored owing to the expansion of the flow cross-sectional area, causing the collapse of bubbles [17,18,19]. Extreme conditions are formed inside the bubble because the internal gas is compressed significantly during bubble collapse [20]. The dissociation of water molecules is activated, thereby resulting in the generation of hydroxyl radicals. Furthermore, a series of reactions such as pyrolysis and mechanical actions, which are the main degradation mechanisms of HC, are initiated in this extreme environment [21]. HC technology has been applied in the treatment of high-concentration complex compound wastewater, both organic and inorganic, combined with oxidants [22,23,24]. Gągol et al. observed that sulfide ions oxidized completely via hydrodynamics aided by H_2_O_2_ within 30 min [25]. Patil et al. applied an HC-based advanced oxidation process to degrade imidacloprid completely under suitable conditions [26]. In the case of Fe^3+^-doped TiO_2_, the degradation rate of RhB reached 91.11% via HC catalytic degradation [27]. HC has been applied to treat antibiotic wastewater, such as sulfadiazine and ciprofloxacin [28,29]; however, there are few studies on TC, and the degradation mechanism of CTC has not been investigated sufficiently.

Hence, the present study was conducted to develop a rapid treatment technology to mitigate CTC in high concentrations. Additionally, the associated mechanism and pathways were determined.

## 2. Materials and Methods

### 2.1. Chemicals

Chlortetracycline hydrochloride (99%), hydrogen peroxide (H_2_O_2_, 30%), oxalic acid (98%), iron sulfate heptahydrate (99%), and sulfuric acid (98%) of analytical grade were obtained from Sinopharm Chemical Reagent Co., Ltd. (Shanghai, China). Formic acid (98%) and acetonitrile (99.8%) of HPLC grade were purchased from S D Fine Chemicals Ltd. (Mumbai, India). All solutions were prepared using a Milli-Q/Milli-Ro Millipore system (Waters Co., Milford, MA, USA). All reagents from the suppliers were used for the experiments without any pretreatment.

### 2.2. Experimental System

Figure 2a shows a schematic diagram of the HC system, which comprises a plunger pump, three control valves, a wastewater tank (1.5 L capacity), a Venturi, a storage tank with a total capacity of 2.0 L, and two pressure sensors with a full scale of 5.0 bar. The geometric details of the Venturi are depicted in Figure 2b. The Venturi has a diameter of 10 mm and a length of 23 mm, with convergent and divergent section lengths of 5.1 mm and 15.9 mm, respectively. Throat angles of the convergent and divergent sections are 40° and 15°, respectively. The diameter and length of the throat are 1.5 mm and 2 mm, respectively. The water in the tank was driven by a 1.1 kW, single-phase 220 V plunger pump (BoQi Pump Co., ZheJiang, China). The holding tank was provided with cooling jacket to allow the circulation of water to maintain the temperature in the range of 25 ± 5 °C. An RS-485 hub was used to obtain data from all sensors; the data were saved in an industrial personal computer for subsequent use. The equipment of the system was connected by DN20 stainless pipes. A certain concentration of CTC solution was filled into the wastewater tank, and water samples were obtained from the storage tank after processing. The inlet pressure was adjusted by controlling the flow via the control valve. Prior to each test, the entire system was fully washed thrice with circulating water for 5 min to eliminate the effects from previous experiments.

### 2.3. Experimental Procedure

Based on previous studies [1,2], all degradation experiments were performed under various conditions at a fixed concentration of 80 mg/L of CTC and 1 L of deionized water. The initial pH of the solution was 6.0, and no matrix components were added. All experiments were performed based on the same solution temperature (25 °C) and duration (30 min).

The degradation of CTC for different inlet pressures over the range of 1.5–3.0 bar was performed for up to 30 min of reaction time in this study. The flow characteristics of the Venturi for each operating pressure are given in Table 1. Under the optimum inlet pressure, the samples were removed to analyze the degradation extent at 5, 10, 20, and 30 min. The degradation extent of CTC at different loadings of H_2_O_2_ (0.5, 1.0, 2.0, 4.0, and 8.0 mM) based on HC was determined at the optimum inlet pressure at 5 and 30 min, separately.

The possible degradation products were determined by comparing the mass spectra before and after the degradation using HC alone and HC combined with the H_2_O_2_ process.

### 2.4. Analytical Methods

An Agilent 1100 High-Performance Liquid Chromatograph system equipped with a lC-2010 variable wavelength ultraviolet/visible detector was used to measure the concentration of CTC. A Hitachi C18 column (4.6 mm × 250 mm, 5 μm) was selected. The mobile phase consisted of acetonitrile and 0.01 mol/L oxalic acid with a volume ratio of 16%:84% at a flow rate of 1 mL/min. The volume of the injected sample was 20 μL, and the column temperature was 25 °C.

A calibration chart was created based on a certain concentration of CTC in the range of 1–200 mg/L to establish a standard operating curve of CTC concentration vs. peak area to calculate the concentration of unknown samples during the experiments. The degradation rate of CTC was calculated using Equation (1).
(1)$$D=\left[1-C_{t}{/C}_{0}\right]\times 100\%,$$

The C_0_ and C_t_ are the initial concentrations of CTC and the concentration at time t (mg/L), respectively, and t is the processing time (min).

The degradation products of CTC were analyzed using an Obitrap Exploris 240 Ultra High Performance Liquid chromatography/mass spectrometer. A Hitachi C18 column (4.6 mm × 250 mm, 5 μm) was selected. The mobile phase consisted of A:acetonitrile + 0.1% formic acid and B:H_2_O + 0.1% formic acid, with a volume ratio of 16%:84% at a flow rate of 0.3 mL/min. The volume of the injected sample was 10 μL, and the column temperature was 25 °C.

All samples to be analyzed were filtered with a 0.22 um filter membrane before being injected.

### 2.5. Ecological Risk for Antibiotics in Water Samples

The ecological risk quotient of individual CTC in aquatic environments is calculated by dividing the measured environmental concentration (MEC) by the predicted no-effect concentration (PNEC) [30].
(2)$$\mathrm{RQ}={\mathrm{MEC}}_{\max}/\mathrm{PNEC},$$

The value of MEC was calculated using the following formula, where D is the degradation of CTC obtained via HC combined with H_2_O_2_ in 30 min, which was set as 93.8% in this study, and C0 is the initial concentration of CTC, which was set to 80 mg/L.
(3)$${\mathrm{MEC}}_{\max}=(1\;-\;D)C_{0},$$

The PNEC was calculated using Equation (3). Chronic toxicity data were preferred for assessing risk, which was predicted using the ECOSAR database (v1.11, USEPA). Meanwhile, the assessment factor, AF, was set as 100 to represent chronic toxicity in the aquatic environment [30].
(4)PNEC=L(E)C50/AF or PNEC=NOEC/AF,

In this study, risk was classified into four levels: insignificant risk (RQ < 0.01), low risk (0.01 < RQ < 0.1), medium risk (0.1 < RQ < 1), and high risk (RQ > 1), which are applied to the ecological risk assessments above.

### 2.6. Energy Consumption Analysis

The ability of the cavitation equipment to produce the desired change based on the electric energy completely used for generating cavitation is represented by cavitational yield (C.Y.), calculated by dividing the degradation rate by the power density.
(5)C.Y.=CTC Degraded/Power density
where CTC Degraded is the amount of CTC removed in the experiment, while power density, in J/L, is represented by Equation (6):(6)Power density=(Pabs·t)/V
where Pabs is the pump absorbed power, t is the time of treatment, and V is the volume of liquid.

## 3. Results

### 3.1. Extent of Degradation of CTC by HC Alone

The results obtained for the effect of inlet pressures on the extent of degradation of CTC over the range of 1.5–3.0 bar within 30 min are depicted in Figure 3a. We observed that the rate of degradation increased with an increase in the inlet pressure from 1.5 to 2.5 bar. The degradation rate increased by 17.3% and 5.8% with inlet pressure increasing from 1.5 to 2.0 bar and from 2.0 to 2.5 bar, respectively, and did not change significantly at a higher inlet pressure (3.0 bar). Considering the wear and tear of the device, an inlet pressure of 2.0 bar was chosen as the optimum pressure for the remaining experiments. The obtained results for using HC at 2.0 bar within 30 min are given in Figure 3b. The degradation rate of CTC was 50.4% for a reaction time of 5 min at an inlet pressure of 2.0 bar and only a slight increase after 5 min.

### 3.2. Extent of Degradation of CTC by HC Combined with Oxidants

The effect of H_2_O_2_ on the degradation extent of CTC based on HC was investigated at the loading of H_2_O_2_ ranging from 0.5 to 8.0 mM. Figure 4 illustrates the effect of H_2_O_2_ concentration on the degradation ratio of CTC after 5 and 30 min of treatment time. The degradation extent of CTC exceeded 85% when HC combined with H_2_O_2_ loading from 2.0 to 8.0 mM was performed within 30 min. The degradation rate increased when the loading of H_2_O_2_ increased from 2.0 to 4.0 mM within 30 min, whereas the degradation extent increased only marginally at 5 min when H_2_O_2_ loading increased from 2.0 to 8.0 mM. Therefore, the best H_2_O_2_ loading for the degradation of CTC based on HC was 2.0 to 4.0 mM.

### 3.3. Product Identification and Pathways

#### 3.3.1. Product Identification and Pathways Based on HC

The UPLC-MS spectra of samples before and after degradation were compared to determine the possible degradation pathways and products for CTC. Six CTC products that are likely to be generated via HC alone were identified, as illustrated in Figure 5. The CTC degradation pathways included ring openings, C–N bond cleavage, demethylation, dehydroxylation, and desaturation. CTC degradation began with the opening of the A carbon ring, followed by demethylation by the cleavage of the C–N bond under the condition of high temperature and pressure or hydroxyl radicals generated by the cavitation effect; subsequently, CTC was transformed to form a tricyclic compound P1. P1 was prone to attack by hydroxyl radicals and pyrolyzation to yield P2 via demethylation. Subsequently, the C ring opened and formed a polyhydroxy compound P3. P3 was dehydroxylated and demethylated, which yielded P4 and P5. Subsequently, P5 underwent desaturation and yielded P6.

#### 3.3.2. Product Identification and Pathways via HC Combined with H_2_O_2_

The mass spectra of each product based on the binary system are shown in Figure 6, and 12 primary products were detected. The P1, P2, P3, P4, P5, and P6 generated were of the same substances as those generated using HC alone. As the binary system proceeded, a series of reactions occurred, such as glycine conjugation and the cleavage of C–C double bonds. When the hydroxyl radicals attacked CTC, the C ring opened, thereby forming a bicyclic compound P7. P7 completed the loss of hydroxyl and methyl groups to form P8, which then resulted in P9 after hydroxyl shedding and C–C double bond cleavage. Meanwhile, under the attack of active substances, CTC completed the ring opening, thereby forming P10 directly under the effect of cavitation. P10 indicated glycine conjugation and yielded P11, followed by ring opening to convert P12. Subsequently, P12 generated P13 through dehydroxylation and demethylation.

### 3.4. Ecological Risk for Antibiotics in Water Samples

The PNEC and RQs referring to the residual CTC remaining in the water for green algae, daphnid, and fish after HC + H_2_O_2_ degradation are shown in Table 2. As shown, the PNEC values of CTC for green algae, daphnid, and fish were 1.9, 0.3, and 8.0 mg/L, respectively. Meanwhile, the RQs of CTC in the water for green algae, daphnid, and fish were 2.6, 16.5, and 0.6, respectively, suggesting that the water samples might pose high ecological risk to green algae and daphnid, and medium risk to fish.

### 3.5. Energy Consumption Analysis

C.Y. is defined as the desired chemical change obtained per unit of power dissipation, which can be used to evaluate and compare the energy consumption of CTC under different conditions. In HC and combination systems, the main energy-consuming source is the plunger pump used for recirculation of the CTC aqueous solution through the cavitation chamber. Table 3 shows C.Y. for different treatment times using HC combined with H_2_O_2_ at an inlet pressure of 2.0 bar. The results show that the C.Y values were 9.0 × 10^−5^ mg/J and 9.5 × 10^−5^ mg/J at 5 min and 30 min, respectively.

## 4. Discussion

### 4.1. Effect of Inlet Pressure

As the pressure increased, the degradation rate increased from 1.5 to 2.5 bar and then did not change with significantly higher pressure. Bagal and Gogate investigated the degradation of 2,4-dinitrophenol at a ranging inlet pressure from 3.0 to 6.0 bar; they discovered that the degradation ratio increased with the inlet pressure until an optimum value of 4.0 bar [31]. Jadhav et al. demonstrated the treatment of imidacloprid using a circular Venturi as a cavitating device combined with oxidants and reported that increasing the inlet pressure from 5 to 15 bar increased the degradation efficiency of imidacloprid and decreased the degradation extent in the operating pressure to 20 bar [32].

During HC, the pyrolysis of molecules inside or near the collapse cavity generated hydroxyl radicals in the system, causing the decomposition of CTC [33]. As the pressure continuously increased, cavitation intensified, thereby resulting in the generation of hydroxyl radicals enhanced by the dissociation of water molecules [12,31]. However, when the pressure exceeded the optimal value, numerous cavities in the Venturi coalesced and formed a cavity cloud, which resulted in choked cavitation [21,34]. Consequently, the strength of the cavity collapse and the probability of a single cavity collapse reduced, thereby resulting in fewer hydroxyl radicals generated, which ultimately reduced the degradation rate of CTC [23].

### 4.2. Effect of Oxidants

The degradation ratio of CTC by HC combined with H_2_O_2_ was similar to that of HC combined with Fenton; however, a higher degradation rate was achieved by the former process compared to using H_2_O_2_ and Fenton alone (Appendix A). Because significant amounts of iron can worsen environmental pollution problems [35,36], experiments were performed without the addition of iron. H_2_O_2_ is a green oxidizing agent that can generate hydroxyl radicals; hence, it is an environmentally friendly element for wastewater treatment [32,37]. In the presence of HC and H_2_O_2_, owing to the high pressure and temperature conditions created by cavitation, the dissociation of H_2_O_2_ and water resulted in numerous hydroxyl radicals, which improved the formation of hydroxyl radicals compared with HC or the H_2_O_2_ process alone [12]. Hence, the combination of HC and H_2_O_2_ resulted in better performances than the application of two processes individually. The following are reactions that may occur when the HC is combined with H_2_O_2_ [32].
(7)$$\overset{\;\mathrm{HC}}{H_{2}O_{2}\;\to \;}2\mathrm{OH},$$
(8)$$\mathrm{OH}\cdot +\mathrm{OH}\cdot \;\to {\;H}_{2}O_{2},$$
(9)$$H_{2}O_{2}+\mathrm{OH}\cdot \;\to {\;\mathrm{HO}}_{2}{\cdot +H}_{2}O,$$
(10)$${\mathrm{HO}}_{2}{\cdot +\mathrm{HO}}_{2}\cdot \to H_{2}O_{2}{+O}_{2},$$
(11)$${\mathrm{OH}\cdot +O}_{2}\;\to {\;\mathrm{HO}}_{2}\cdot +O,$$

As H_2_O_2_ loading increased, H_2_O_2_ was constantly dissociated under cavitation, resulting in an enhanced formation of hydroxyl radicals, as shown in Equation (7), which remarkably promoted the degradation of CTC [38]. Nevertheless, excessive H_2_O_2_ loading beyond the optimal value could cause recombination and scavenging reactions [21,39], as shown in Equations (8)–(11), which resulted in a decrease in the concentration of hydroxyl radicals and the generation of some other radicals with less oxidative power, whereas the degradation ratio of CTC did not further increase significantly. Wang et al. obtained a similar trend in the degradation of azo dyes via a process that combined HC and H_2_O_2_ [39].

### 4.3. Possible Mechanisms of CTC Degradation

Chemical structures of the products and degradation pathways of CTC were suggested based on the molecular structure of CTC and the mass of the products. In both HC and HC combined with H_2_O_2_ processes, hydroxyl radicals attacked the functional groups in zone 1 and zone 2 connected by the A ring, including the cleavage of the C–O double bond at C1, the elimination of the functional groups at C2 and C3, and the breaking of the C–N bond [6,40]. A ring completed the opening reaction caused by the C2–C3 double bond cleavage. Demethylation occurred at C6, resulting in the loss of the methyl group [41], followed by the breaking of C5a–C6 and C11–C11a of the C ring, which caused the opening of the C ring, followed by its oxidization to form a polyhydroxy monocyclic compound adding two hydroxyl groups.

In the HC process combined with H_2_O_2_, OH radicals can directly attack the C5a–C6 and C11–C11a bonds on the C ring to form bicyclic compounds. In addition, hydroxyl radicals can directly attack the B ring, including the C12–C12a bond, to form a hydroxyl group and a keto group at C12 with *m*/*z* 284.08, as discovered in a previous study [42]. Subsequently, the hydroxyl radicals attack the CTC, resulting in smaller organic fragments without rings. HC combined with H_2_O_2_ generated more degradation products and a more complicated mechanism compared with HC alone. Oxidation of other oxidizing substances such as H_2_O_2_ and free radicals (HO_2_·) produced by HC combined with H_2_O_2_ may be the reason for the more complicated mechanism in the binary system.

The purpose of this study was to develop a treatment technology for wastewater from CTC pharmaceutical factories. We prepared high-concentration CTCs for the experiments. Although the degradation ratio exceeded 90%, 5.0 mg/L of residue remained. Hence, the water sample posed ecological risks and adverse impacts on green algae, daphnid, and fish. Therefore, this method can only be used as a primary treatment. Other treatment processes such as the chemical and biological methods should be combined as the secondary treatment to treat high-CTC-concentration wastewater from pharmaceutical factory sites [43,44].

The limitations of this study are as follows. Although mass spectrometry revealed the degradation products and possible pathways for understanding the degradation mechanism of CTC, the product structure was speculative and uncertain. Only the parent compound of CTC was considered in the ecological risk assessment; the bioavailability of the degradation products should also be considered. HC is still in the theoretical research stage of mechanism and structural optimization of Venturi, and our study is a pilot-scale experiment; thus, more efforts are necessary to realize the application of HC in reality.

## 5. Conclusions

In this study, we examined the extent of CTC degradation using HC operated alone and combined with oxidants. The degradation ratio was 50.4% after 5 min using HC alone, with an optimal inlet of 2.0 bar. The degradation ratio of CTC by HC combined with H_2_O_2_ was similar to that of HC combined with Fenton, while the former had no addition of iron and less pollution to the environment. The C.Y. values of using HC combined with H_2_O_2_ were 9.0 × 10^−5^ mg/J and 9.5 × 10^−5^ mg/J at an inlet pressure of 2.0 bar at 5 min and 30 min, respectively. Based on the structure and properties of CTC, possible degradation products and reaction mechanisms of HC and HC combined with H_2_O_2_ were proposed. CTC completed the ring openings, C–N bond cleavage, demethylation, dehydroxylation, and desaturation during HC, and a series of additional reactions, such as glycine conjugation and C–C double bond cleavage, occurred in HC combined with H_2_O_2_. HC combined with H_2_O_2_ is a rapid and effective preliminary treatment technology for high concentrations of CTC. It can reduce the danger of the direct discharge of antibiotics to the environment and is feasible for the primary treatment of pharmaceutical factories.

## Figures and Tables

**Figure 1 ijerph-19-04167-f001:**
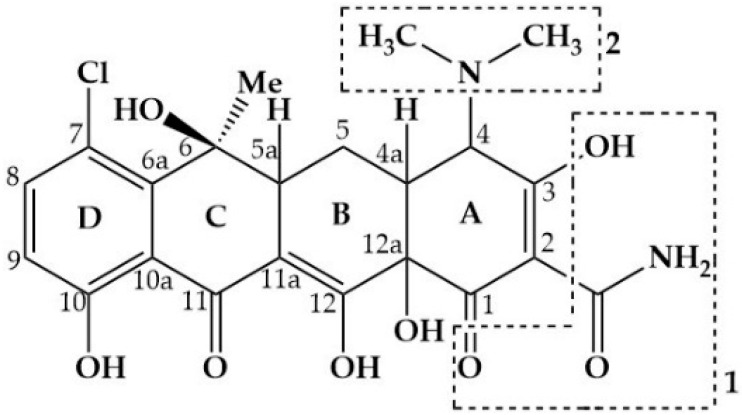
Chemical structure of CTC.

**Figure 2 ijerph-19-04167-f002:**
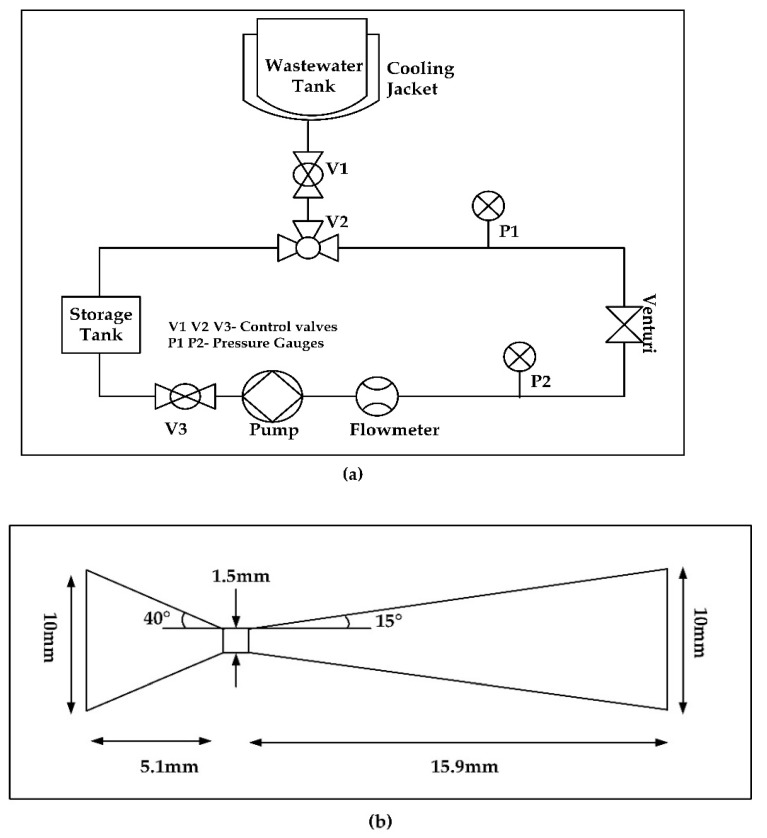
Schematic representation of HC set-up. (**a**) Schematic diagram of the HC system. (**b**) Geometric specifications of a Venturi used as a cavitating device.

**Figure 3 ijerph-19-04167-f003:**
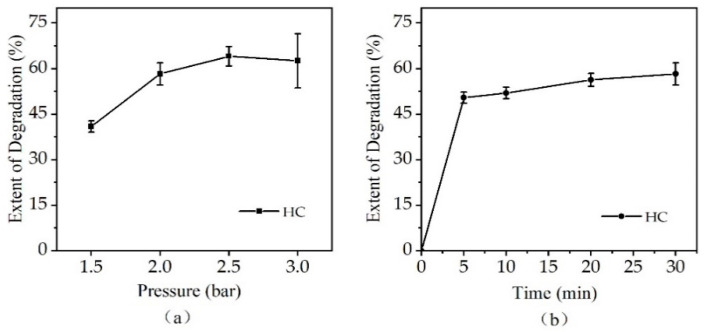
Effect of inlet pressure on extent of degradation of CTC by HC within 30 min: (**a**) effect of inlet pressure within 30 min; (**b**) extent of degradation of CTC by HC with 2.0 bar.

**Figure 4 ijerph-19-04167-f004:**
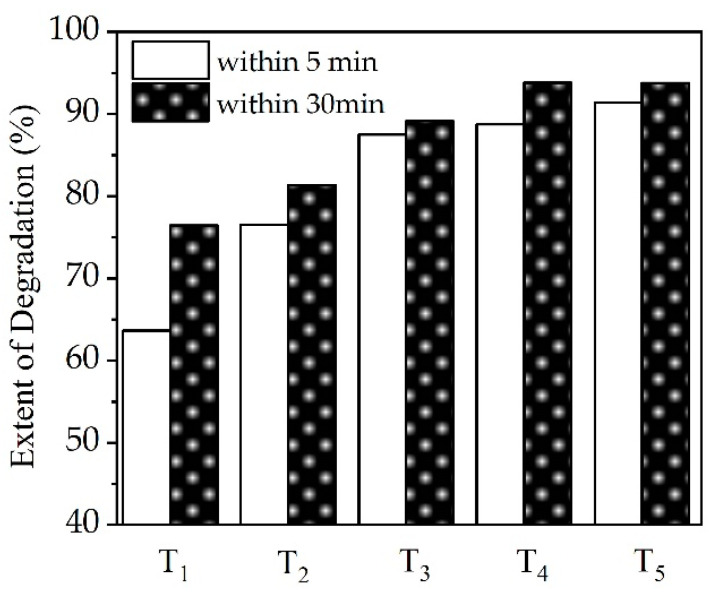
Effect of H_2_O_2_ on extent of degradation of CTC for the combination of HC with H_2_O_2_ at 5 min and 30 min. T1: HC + 0.5 mM H_2_O_2_, T2: HC + 1.0 mM H_2_O_2_, T3: HC + 2.0 mM H_2_O_2_, T4: HC + 4.0 mM H_2_O_2_, T5: HC + 8.0 mM H_2_O_2_.

**Figure 5 ijerph-19-04167-f005:**
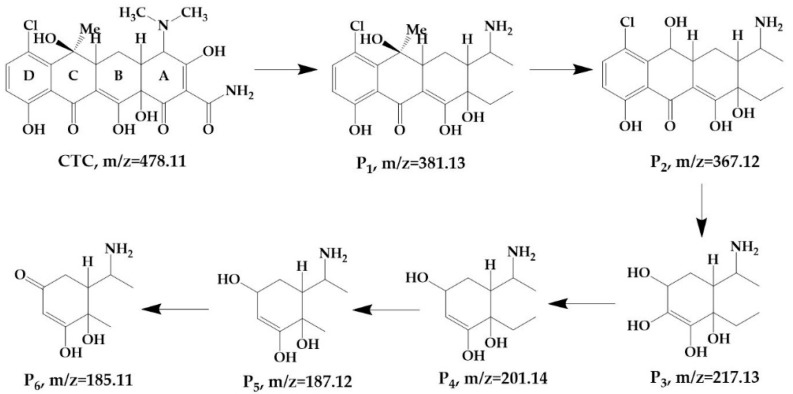
Possible pathways and degradation products of CTC by using HC.

**Figure 6 ijerph-19-04167-f006:**
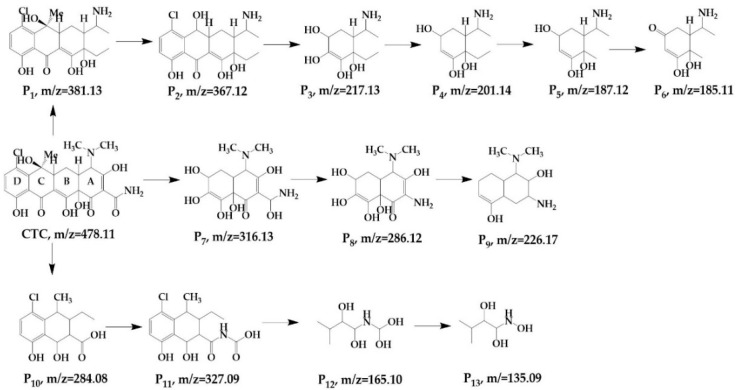
Possible pathways and degradation products of CTC by HC combined with H_2_O_2_.

**Table 1 ijerph-19-04167-t001:** Flow characteristics of Venturi.

Inlet Pressure (bar)	Flow Rate (LPH)	Number of Entire Fluids Passes per Minute
1.5	180	3
2.0	240	4
2.5	300	5
3.0	360	6

**Table 2 ijerph-19-04167-t002:** The PNEC and RQs of CTC for green algae, daphnid, and fish in water.

Organism	PNEC (mg/L)	RQ
Green algae	1.9	2.6
Daphnid	0.3	16.5
Fish	8.0	0.6

**Table 3 ijerph-19-04167-t003:** C.Y. for different treatment times using HC combined with H_2_O_2_.

Treatment Time (min)	Degradation Rate (%)	CTC Degraded (mg/L)	Power Density (kJ/L)	C. Y. (mg/J)
5	88.7	71.0	792	9.0 × 10^−5^
10	88.7	71.0	792	9.0 × 10^−5^
20	91.0	72.8	792	9.2 × 10^−5^
30	93.8	75.0	792	9.5 × 10^−5^

## Data Availability

All data generated or analyzed during this study are available from the corresponding author upon reasonable request.

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
