# Peer review of "Rapid Degradation of Chlortetracycline Using Hydrodynamic Cavitation with Hydrogen Peroxide"

_ijerph, 2022, doi:10.3390/ijerph19074167_

Round 1
Reviewer 1 Report
In this study, the authors tried to accelerate the degradation of the antibiotic chlortetracycline (CTC) by adding hydrogen peroxide into the hydrodynamic cavitation (HC). The degradation mechanism of CTC was investigated.
As the present authors referred some previous studies, it has been known that HC can be used to degrade organic materials in water via the generation of hydroxyl radicals. It is also known that the combination of HC and the hydrogen peroxide increases the amount of generated hydroxyl radicals and the rate of degradation of organic materials. It is easily expected that this water purification method (i.e., the combination of HC and H2O2) will improve the degradation rate of CTC. Therefore, the potential value of this study seems to be how CTC can be degraded into harmless materials. However, this study concludes that CTC and its intermediates, which have adverse effects on aquatic organisms, remained in the water after the treatment proposed here, thus re-treatment was required. This is not beneficial from either an environmental or an economic point of view.
In addition, there are inconsistencies in the examination of the decomposition mechanism. The intermediate compounds during the decomposition of CTC by HC were different from those by HC and H2O2. According to the authors, the reason for the difference in the decomposition mechanism between the reaction with HC only and that with HC plus H2O2 is the concentration of hydroxyl radicals. This means that both the active species in the two reaction conditions are hydroxyl radicals. However, if the active species is the same between the two reaction conditions, the degradation mechanism is expected to be the same regardless of the concentration. Therefore, it is unreasonable to assume that difference in the amount of generated hydroxyl radicals is the reason why the decomposition mechanism differs between HC only and HC plus H2O2.
Form the above-mentioned reasons, the reviewer does not recommend this article for publication.
Author Response
Response to Reviewer 1 Comments
Thank you very much for your careful review and suggestions with regard to our manuscript. Those comments are helpful for authors to revise and improve our paper. We have studied comments carefully and tried our best to revise and improve the manuscript.
Point 1:In this study, the authors tried to accelerate the degradation of the antibiotic chlortetracycline (CTC) by adding hydrogen peroxide into the hydrodynamic cavitation (HC). The degradation mechanism of CTC was investigated.
As the present authors referred some previous studies, it has been known that HC can be used to degrade organic materials in water via the generation of hydroxyl radicals. It is also known that the combination of HC and the hydrogen peroxide increases the amount of generated hydroxyl radicals and the rate of degradation of organic materials. It is easily expected that this water purification method (i.e., the combination of HC and H2O2) will improve the degradation rate of CTC. Therefore, the potential value of this study seems to be how CTC can be degraded into harmless materials. However, this study concludes that CTC and its intermediates, which have adverse effects on aquatic organisms, remained in the water after the treatment proposed here, thus re-treatment was required. This is not beneficial from either an environmental or an economic point of view.
Response 1: This study was to degrade the high concentration of CTC in water. Currently, there is not any single or combined treatment techniques could treat high-concentration of CTC in pharmaceutical wastewater and discharge them into environment directly. Thus, it is significant to explore a new technique at present. In addition, to some extent, any treatment process could produce intermediates and HC also caused intermediates. While in some degradation pathway in our study, the intermediates could be eventually changed into CO2 and H2O. Recently, Ji Wang et al. reached 74.2% degradation rate of CTC in microbial fuel cells in 7days [1]. Rama Pulicharla et al. combined laccase treatment with ultrasonication process showed 87% of CTC removal in 48 h [2]. Liang, Shuang et al. observed around 80% of CTC was degraded in 60 min by Pt/CuO-NS-2 [3]. The degradation rate of this study could reach 93.8% in 30 min using combined processes, which was relatively high with shorter time, having positive meaning.
Point 2:In addition, there are inconsistencies in the examination of the decomposition mechanism. The intermediate compounds during the decomposition of CTC by HC were different from those by HC and H2O2. According to the authors, the reason for the difference in the decomposition mechanism between the reaction with HC only and that with HC plus H2O2 is the concentration of hydroxyl radicals. This means that both the active species in the two reaction conditions are hydroxyl radicals. However, if the active species is the same between the two reaction conditions, the degradation mechanism is expected to be the same regardless of the concentration. Therefore, it is unreasonable to assume that difference in the amount of generated hydroxyl radicals is the reason why the decomposition mechanism differs between HC only and HC plus H2O2.
Response 2: As we shown in original manuscript in line 281-284, the degradation system of HC and H2O2 could not only generate OH·, but also produces HO2·, which also has oxidation effect [4]. In addition, H2O2 also has oxidation ability. Therefore, not only hydroxyl radicals but also other oxygen radicals play a role in the combined treatment processes. We have modified it in line 310-312.
References
- Wang, J.; Zhou, B.; Ge, R.; Song, T.-s.; Yu, J.;Xie, J., Degradation characterization and pathway analysis of chlortetracycline and oxytetracycline in a microbial fuel cell. RSC Advances, 2018. 8(50): p. 28613-28624.http://dx.doi.org.10.1039/c8ra04904a.
- Pulicharla, R.; Das, R.K.; Brar, S.K.; Drogui, P.;Surampalli, R.Y., Degradation kinetics of chlortetracycline in wastewater using ultrasonication assisted laccase. Chemical Engineering Journal, 2018. 347: p. 828-835.http://dx.doi.org.10.1016/j.cej.2018.04.162.
- Liang, S.; Zhou, Y.M.; Wu, W.T.; Zhang, Y.W.; Cai, Z.L.;Pan, J., Preparation of porous CuO nanosheet-liked structure (CuO-NS) using C3N4 template with enhanced visible-light photoactivity in degradation of chlortetracycline. Journal of Photochemistry and Photobiology a-Chemistry, 2017. 346: p. 168-176.http://dx.doi.org.10.1016/j.jphotochem.2017.06.005.
- Raut-Jadhav, S.; Saharan, V.K.; Pinjari, D.; Sonawane, S.; Saini, D.;Pandit, A., Synergetic effect of combination of AOP's (hydrodynamic cavitation and H(2)O(2)) on the degradation of neonicotinoid class of insecticide. J Hazard Mater, 2013. 261: p. 139-47.http://dx.doi.org.10.1016/j.jhazmat.2013.07.012.

Reviewer 2 Report
Paper presents interesting approach of chlortetracycline degradation by combining hydrodynamic cavitation and hydrogen peroxide. The methods seems to be very effective. In my opinion one thing is missing. Namely, the Figure 2, which Authors placed, is of illustrative purposes only. Authors should present the more detailed graphical interpretation of device or apparatus used to better show of the measurements.
Author Response
Response to Reviewer 2 Comments
Thank you very much for your kindly comments on our manuscript. There is no doubt that the comments are valuable and very helpful for revising and improving our manuscript.
Point:Paper presents interesting approach of chlortetracycline degradation by combining hydrodynamic cavitation and hydrogen peroxide. The methods seems to be very effective. In my opinion one thing is missing. Namely, the Figure 2, which Authors placed, is of illustrative purposes only. Authors should present the more detailed graphical interpretation of device or apparatus used to better show of the measurements.
Response: Additional details of venturi data, including divergence angle, contraction angle, throat diameter, etc. have been supplemented in Figure 2 (b) and mentioned in line 87-91.

Round 2
Reviewer 1 Report
Now I believe the manuscript has been sufficiently improved to warrant publication in IJERPH.